

# Characterization of aerosol properties at Cyprus, focusing on cloud condensation nuclei and ice nucleating particles

Xianda Gong[1], Heike Wex[1], Thomas Müller[1], Alfred Wiedensohler[1], Kristina Höhler[2], Konrad Kandler[3], Nan Ma[1], Barbara Dietel[2], Thea Schiebel[2], Ottmar Möhler[2], and Frank Stratmann[1]

[1]Experimental Aerosol and Cloud Microphysics Department, Leibniz Institute for Tropospheric Research, Leipzig, Germany
[2]Institute for Meteorology and Climate Research – Atmospheric Aerosol Research, Karlsruhe Institute of Technology, Karlsruhe, Germany
[3]Institute for Applied Geosciences, Technical University Darmstadt, Darmstadt, Germany

**Correspondence:** Xianda Gong (gong@tropos.de)

**Abstract.** As part of the A-LIFE (**A**bsorbing aerosol layers in a changing climate: aging, **life**time and dynamics) campaign, ground-based measurements were carried out in Paphos, Cyprus, for characterizing the abundance, properties and sources of aerosol particles in general, and cloud condensation nuclei (CCN) and ice nucleating particles (INP), in particular. New particle formation (NPF) events with subsequent growth of the particles into the CCN size range were observed. Aitken mode particles featured $\kappa$ values of 0.21 to 0.29, indicating the presence of organic materials. Accumulation mode particles featured a higher hygroscopicity parameter, with a median $\kappa$ value of 0.57, suggesting the presence of sulfate. A clear downward trend of $\kappa$ with increasing supersaturation and decreasing $d_{\text{crit}}$ was found. Super-micron particles originated mainly from sea spray aerosol (SSA) and partly from mineral dust.

INP concentrations ($N_{\text{INP}}$) were measured in the temperature range from $-6.5$ to $-26.5\,°C$, using two freezing array type instruments. $N_{\text{INP}}$ at a particular temperature span around 1 order of magnitude below $-20\,°C$, and about 2 orders of magnitude at warmer temperatures (T$> -18\,°C$). Few samples showed elevated concentrations at temperatures $> -15\,°C$, which suggests a significant contribution of biological particles to the INP population, which possibly could originate from Cyprus. Both measured temperature spectra and $N_{\text{INP}}$ probability density functions (PDFs) indicate that the observed INP (ice active in the temperature range between $-15$ and $-20\,°C$) mainly originate from long-range transport. There was no correlation between $N_{\text{INP}}$ and particle number concentration in the size range $>500$ nm ($N_{>500\text{nm}}$). Parameterizations based on $N_{>500\text{nm}}$ were found to overestimate $N_{\text{INP}}$ by about 1 to 2 orders of magnitude. There was also no correlation between $N_{\text{INP}}$ and particle surface area concentration. The ice active surface site density ($n_{\text{s}}$) for the anthropogenically polluted aerosol encountered in this study is about 1 to 3 orders of magnitude lower than the $n_{\text{s}}$ found for dust aerosol particles in previous studies. This suggests that observed $N_{\text{INP}}$-PDFs as those derived here could be a better choice for modelling $N_{\text{INP}}$ if the aerosol particle composition is unknown or uncertain.




# 1  Introduction

The Mediterranean region is one of the hot-spot areas of the globe being severely threatened by climate change (Giorgi and Lionello, 2008) with the direct and indirect effects of aerosol particles therein still remaining unclear. The Mediterranean region is rich in a variety of aerosols (fuel combustion, biomass burning, secondary biogenic, sea spray and mineral dust aerosols) from both continental and marine sources (Chester et al., 1993; Piazzola and Despiau, 1997; Lelieveld et al., 2002). The sensitivity of this region, together with the large number of influencing factors, makes it a difficult task to understand all ongoing processes and their interconnections. This, however, is needed in order to better be able to protect the region or mitigate upcoming changes. Our goal in this frame is to better understand the varied aerosol that occurs in this region. In the next paragraphs, we will start by giving an overview on what is known about the Mediterranean aerosol.

Regarding anthropogenic sources of aerosol particles, Sciare et al. (2003) found that the major contributions in the eastern Mediterranean were from Turkey and central Europe. Central Europe was identified as the major source of black carbon over the eastern Mediterranean. In the Po Valley, which is in the western Mediterranean, but which we still consider here, due to the comparable climatic conditions, Sandrini et al. (2016) found that particles in the size range from 50 to 140 nm were mainly from traffic emissions. The photochemical oxidation of inorganic and organic gaseous precursors was identified as the important mechanism of secondary aerosol formation, which caused the accumulation mode (420-1200 nm) aerosol particles to be constituted mainly of ammonium nitrate, organic carbon and sulfate. Bougiatioti et al. (2013) found that organic carbon and element carbon concentrations made up 2/3 for the $PM_1$, with organic carbon being mostly secondary and therefore highly oxidized and water-soluble to a great extent.

Biomass burning is another important anthropogenic aerosol source over the Mediterranean, and it was mainly observed in the driest months of the year, July and August (Pace et al., 2005). Long-term observations of absorbing aerosol particles have clearly shown that they originated from agriculture waste burning (post-harvest wheat residual) in the countries surrounding the Black Sea (Sciare et al., 2008). Bougiatioti et al. (2016) examined in the eastern Mediterranean potential cloud condensation nuclei (CCN) and hygroscopicity properties and found that an increased organic content in the aerosol particles decreased the values of the hygroscopicity parameter $\kappa$ for all particle sizes. Furthermore, they observed CCN concentrations ($N_{CCN}$) to be enhanced by a factor from 1.6 to 2.5 during biomass burning plumes compared to background conditions.

Natural aerosol particles such as mineral dust and sea salt are however the major contributing factors to particle mass in the Mediterranean (Rodríguez et al., 2002). Mineral dust particles from the Sahara were regularly observed at different locations across the Mediterranean. A record-breaking dust storm originating from desert regions in northern Syria and Iraq occurred over the eastern Mediterranean in September 2015. The $PM_{10}$ concentrations were close to 8000 $\mu$g m$^{-3}$ and the observed meteorological optical range (MOR) was reduced to 300-750 m (Mamouri et al., 2016). By using the Weather Research and Forecasting model in a Sahara outflow region, Smoydzin et al. (2012) found that the presence of mineral dust can enhance the CCN concentration and formation of ice crystals.

Sea spray aerosols (SSA) are another main natural aerosol type observed in the Mediterranean. Claeys et al. (2017) found that primary marine aerosols mass concentration reached up to 6.5 $\mu$g m$^{-3}$, representing more than 40% of the total $PM_{10}$



mass in the western Mediterranean. Salameh et al. (2007) reported AOD around 0.15-0.20 (at 865 nm) within a SSA plume during strong wind events with wind speeds up to 18 m s$^{-1}$.

Clouds in the atmosphere form when water vapor condenses on aerosol particles that serve as CCN. Clouds in the atmosphere glaciate at temperatures above $-38\,^{\circ}$C if droplet freezing in initiated by aerosol particles called ice nucleating particles (INP),
or at temperatures below $-38\,^{\circ}$C also through homogeneous freezing (without INP) (Pruppacher and Klett, 2010). Therefore, a change in atmospheric aerosol particles, especially CCN and INP, is bound to impact cloud properties, precipitation, and cloud radiative effects (Fan et al., 2016). Even though clouds are omnipresent in the Earth's atmosphere, and constitute an important role in regulating the radiative budget of the planet, the response of clouds to climate change remains highly uncertain, in particular with regard to aerosol-cloud interactions and feedback mechanisms.

In-situ observations of CCN on Crete were reported by Kalivitis et al. (2015), highlighting new particle formation (NPF) as a source of CCN. At Finokalia, Crete, Bougiatioti et al. (2011) found that air masses originating from central eastern Europe tend to be associated with higher $N_{CCN}$, and slightly lower hygroscopicity ($\kappa = 0.18$), than other air masses.

Seldomly, measurements of INP have been carried out in the Mediterranean. Excluding situations characterized by high altitude transport of dust plumes, Rinaldi et al. (2017) found that at a measurement station in the Po valley basin, INP number
concentration ($N_{INP}$) was roughly double that of what they observed at the top of an Apeninne mountain. Schrod et al. (2017) found that mineral dust, or a constituent related to dust, was a major contributor to INP on Cyprus. However, due to Sahara dust plumes travelling at several kilometer altitude, $N_{INP}$ at higher altitudes was 10 times higher than at ground level (height $\sim$700 m).

As outlined above, the aerosol in the Mediterranean region represents a complex mixture of primary and secondary aerosol
particles from both natural and anthropogenic sources, with these sources being non-uniformly distributed across the greater Mediterranean region. Most regional and global climate simulations have investigated impacts of global warming on the Mediterranean climate without detailed considerations of possible radiative influences and climatic feedback from different types of Mediterranean aerosols (Mallet et al., 2016). Besides, to the best of our knowledge, seldom studies paid attention to the CCN and INP simultaneously, which both have an effect on climate. The aim of this study is to provide a quantitative
understanding concerning the abundance, properties and source of CCN and INP in the eastern Mediterranean.

## 2 Experimental

### 2.1 Sampling site and campaign setup

Measurements were performed from 2 to 30 of April 2017, on the island of Cyprus, as part of the A-LIFE (**A**bsorbing aerosol layers in a changing climate: aging, **life**time and dynamics) project, which had the purpose to investigate properties of absorbing
aerosols during their atmospheric lifetime, and their distribution throughout the tropospheric column. Cyprus, an island located in the eastern Mediterranean region, is approximately 100 km south of the Turkish mainland, 100 km west of the Syrian, and 300 km north of the Egyptian coast. This geographical location makes Cyprus an unique spot in the eastern Mediterranean Sea, where different and complex aerosol mixtures occur. On one hand, the Sahara Desert in the southwest, and the desert of the





Arabian Peninsula in the southeast, favor a regular occurrence of mineral-dust-rich air masses. One the other hand, Cyprus is influenced by anthropogenic emissions from southeastern Europe, as well as the Middle East, and of course, local pollution. This exposure to diverse air masses makes Cyprus an ideal place for investigating the abundance and properties of climate relevant aerosol particles in general, and CCN and INP, in particular. As shown in Fig. 1, the measurement site was located in

Paphos, Cyprus ($34°43'$ N, $32°29'$ E). The measurements took place at the side of a fairly calm coastal highway, facing the Mediterranean Sea. On the northeastern side of the measurement site, 1 km away, is the Paphos International Airport.

The instrumental setup used for these investigations is shown in Fig. S1. An aerosol $PM_{10}$ inlet, employed to remove particles larger than 10 $\mu$m in aerodynamic diameter, was installed on top of a measurement container. Downstream of the $PM_{10}$ inlet, a vertical tube (inner diameter of 1.65 cm), and a diffusion dryer (130 cm), were arranged before the aerosol was lead into the

measurement container. The diffusion dryer was installed vertically to avoid gravitational losses of larger particles. Downstream of the dryer and inside the container, the sampled aerosol was split to supply the aerosol to various instruments, measuring particle number size distribution (PNSD), number concentration, as well as hygroscopic and optical (not discussed in this paper) properties.

For the measurement of $N_{INP}$, two different filter-based sampling systems were utilized. For one set of samples, total sus-

pended particles were collected with a flow rate of $\sim$10 L min$^{-1}$. For a second set of samples, a separate $PM_{10}$ inlet was used as inlet, and an air flow of $\sim$15 L min$^{-1}$ was sampled onto the filters. No dryer was arranged in the filter sampling system.

The CCN hygroscopicity was derived from $N_{CCN}$ combined with the PNSD. INP freezing behavior and $N_{INP}$ were determined by filter sampling and off-line analysis using freezing array type instruments. In the following, we will briefly introduce the different measurement techniques applied in this study, including calibrations, measurements and data processing.

And lastly, to get additional information on the presence of super-micron particles, depositing aerosol particle were collected at ambient conditions outside of the measurement container.

### 2.2 Particle number size distribution

PNSDs were measured in the size range from 10 nm to 10 $\mu$m using a TROPOS-type MPSS (Mobility Particle Size Spectrometer) (Wiedensohler et al., 2012), and an APS (Aerodynamic Particle Sizer, model 3321, TSI Inc., St. Paul, MN, USA).

For the multiple charge correction (Wiedensohler, 1988) of the MPSS data, the APS data was accounted for in the inversion of the measured PNSD (Pfeifer et al., 2016). The combined PNSD is then given on the base of the volume equivalent particle diameter. More details about the combined MPSS and APS PNSD can be found in Schladitz et al. (2011). Size-dependent particle losses due to diffusion, deposition and sedimentation within the inlet were corrected for utilizing the empirical particle loss calculator (von der Weiden et al., 2009), as shown in Fig. S2. Total particle number concentrations ($N_{total}$) were calculated

from the measured PNSDs and the size-dependent particle losses. The calibration of the MPSS before, during and after the intensive field study was done following the recommendations given in Wiedensohler et al. (2018).





## 2.3 Cloud condensation nuclei

$N_{\mathrm{CCN}}$ was measured using a Cloud Condensation Nuclei counter (CCNc, Droplet Measurement Technologies (DMT), Boulder, USA). The CCNc is a cylindrical continuous-flow thermal-gradient diffusion chamber, establishing a constant streamwise temperature gradient to adjust a quasi constant centerline supersaturation. The sampled aerosol particles are guided within a
sheath flow through this chamber and can become activated into droplets, depending on the supersaturation and the particles' ability to act as CCN. The details of the CCNc are described in Roberts and Nenes (2005).

During our study, the supersaturation was varied from ∼0.08 % to ∼0.77 % at a constant total flow rate of 0.5 L min$^{-1}$. To assure stable column temperatures, the first 5 minutes and the last 30 seconds of the 12-minute long measurement at each supersaturation, were excluded from the data analysis. The remaining data points were averaged. A supersaturation calibra-
tion (following the protocol by Gysel and Stratmann, 2013) was done at the cloud laboratory of the Leibniz Institute for Tropospheric Research (TROPOS) prior to and after the measurement campaign, to determine the relationship between the temperature gradient along the column and the effective supersaturation. Calibrated supersaturation set-points were 0.08 %, 0.19 %, 0.31 %, 0.54 % and 0.77 %. These calibrated values were used for further calculations.

According to Köhler theory (Köhler, 1936), whether or not a particle can act as a CCN depends on its dry size, chemical
composition and the maximum supersaturation it encounters. Petters and Kreidenweis (2007) presented a method to describe the water activity term in the Köhler equation by utilizing the hygroscopicity parameter $\kappa$. The $\kappa$ values reported in this study were calculated as follows, assuming the surface tension of the examined solution droplets $\sigma_{s/\alpha}$ to be that of pure water:

$$\kappa = \frac{4A^3}{27d_{\mathrm{crit}}^3 \ln^2 S} \tag{1}$$

with

$$A = \frac{4\sigma_{s/\alpha}M_{\mathrm{w}}}{RT\rho_{\mathrm{w}}} \tag{2}$$

where $d_{\mathrm{crit}}$ is the critical diameter above which all particles activate into cloud droplets for a given supersaturation. $M_{\mathrm{w}}$ and $\rho_{\mathrm{w}}$ are the molar mass and density of water, while $R$ and $T$ are the ideal gas constant and the absolute temperature, respectively. To derive $d_{\mathrm{crit}}$, simultaneously measured $N_{\mathrm{CCN}}$ and PNSD are used. Thereto, it is assumed that all particles in the neighborhood of a given particle diameter have a similar $\kappa$, meaning that the aerosol particles are internally mixed. At a given supersaturation,
a particle can be activated to a droplet once its dry size is equal to or larger than $d_{\mathrm{crit}}$. Therefore, $d_{\mathrm{crit}}$ is the diameter at which $N_{\mathrm{CCN}}$ is equal to the value of cumulative particle number concentration, determined via integration from the upper towards the lower end of the PNSD. Hygroscopicity $\kappa$ can be calculated with $d_{\mathrm{crit}}$ and the corresponding supersaturation, based on Eq.(1). Note that the particle losses inside the CCNc (discussed in Rose et al., 2008) are also considered before $\kappa$ is calculated. More details about the correction method and data processing can be found in previous literature (Kristensen et al., 2016; Herenz
et al., 2018).



## 2.4 Ice nucleating particles

We used two setups to sample airborne particles for further analysis. With the first setup, particles were collected on 200 nm pore size polycarbonate filters (Nuclepore Track-Etch Membrane, Whatman) with ∼20 hours time resolution and a flow rate of ∼10 L min$^{-1}$. As shown in Fig. S1, we used a computer-based system to switch between filters based on wind directions.

Two sectors were distinguished, i.e., the ocean sector comprising wind directions from 120 to 240 degree, and the land sector, covering the remaining directions. During the campaign, we collected 4 filters with air from the ocean sector, 17 from the land sector, and 2 blind filter samples in total. All of the filters were stored at −18 °C on Cyprus and cooled below 0 °C during transportation. The start and end times of sampling, flow rates, duration and total sample volumes, are shown in Tab. S1. These filters were transported to TROPOS for analysis. At TROPOS, all filters were stored at −18 °C until they were prepared for the

measurement. Each filter was immersed into 1 mL ultrapure water (Type 1, Millipore) and shaken for 25 minutes to wash off the particles. The resulting water samples were characterized with the Leipzig Ice Nucleation Array (LINA). LINA is based on the freezing array technique and follows the design described in Budke and Koop (2015). Briefly, 90 droplets with a volume of 1 $\mu$L are pipetted from the water samples onto a thin hydrophobic glass slide, with the droplets being separated from each other inside individual compartments. The compartments are sealed at the top with another glass slide, to minimize evaporation of

the droplets and to prevent ice seeding from neighbouring droplets. The bottom glass slide is cooled with a Peltier element with a cooling rate of 1 K min$^{-1}$. A camera takes pictures every 6 seconds, corresponding to a temperature resolution of 0.1 K. The number of frozen versus unforzen droplets was derived automatically. More details concerning the experimental parameters and temperature calibration of LINA can be found in Chen et al. (2018).

For the second filter-based sampling system, 200 nm pore size polycarbonate filters (Nuclepore Track-Etch Membrane,

Whatman) were pre-treated with 10% $H_2O_2$ solution, washed with particle free ultrapure water and dried prior to insertion into the filter holder. Daily filter samples with an air flow rate of ∼15 L min$^{-1}$ for ∼8 hours were taken. In total 25 day time and 2 blind filter samples were collected. All of the filters were stored at −18 °C in Cyprus and cooled below 0 °C during transportation. The start and end times of sampling, flow rates and duration are shown in Tab. S2. These filters were transported to the Karlsruhe Institute of Technology (KIT) for analysis with the Ice Nucleation SpEctrometer of the Karlsruhe

Institute of Technology (INSEKT). INSEKT is a droplet freezing device, the design of which was inspired by the Colorado State University Ice Spectrometer (Hiranuma et al., 2015). For the analysis, each filter was washed with 8 mL ultrapure water, which had been passed through a 0.1 $\mu$m filter (Nuclepore Track-Etch Membrane, Whatman). 50 $\mu$L samples of the resulting suspension/solution were pipetted into 24 to 36 sections of two 96-well PCR trays. Other wells of the trays were filled with 15- and 225-fold (and for some samples also 3375-fold) dilutions of the filter washing water. Also, in each experiment at least

24 wells were filled with pure and particle free water, to be able to account for impurities resulting from the washing water and PCR tray surfaces. The PCR trays were then placed into aluminum cooling blocks. Those blocks have been customized by drilling channels into the bulk aluminum, through which the cooling agent thermostated by means of an external chiller (LAUDA PROLINE RP 855) is directed. The temperature of the cooling agent is then lowered by 0.33 K min$^{-1}$ and monitored





by eight calibrated temperature sensors inserted into the aluminum blocks. The number of frozen versus unfrozen wells was derived visually in 0.5 K steps.

For both measurement systems, the cumulative concentration of INP per air volume as a function of temperature can be calculated based on Vali (1971):

$$N_{\mathrm{INP}}(\theta) = \frac{\ln N_{\mathrm{t}} - \ln N(\theta)}{V} \tag{3}$$

where $N_{\mathrm{t}}$ is the number of droplets/wells and $N(\theta)$ is the number of unfrozen droplets/wells at temperature $\theta$. $V$ means the volume (at 0 °C and 1013 hPa) of air distributed into each droplet/well.

The background freezing signal of ultrapure water and water samples resulting from washing of blind filters is determined for the two sampling systems as well. Measured $N_{\mathrm{INP}}$ is corrected by subtracting the background concentrations determined for the blind filters and the ultrapure water.

Due to the usually small number (order of tens and lower per examined droplet/well) of INP present in the washing water, and the limited number of droplets/wells considered in our measurements, statical errors need to be considered in the data evaluation. Therefore, confidence intervals for the frozen fraction ($f_{\mathrm{ice}}$) were determined using the method suggested by Agresti and Coull (1998). More details about the background subtraction and measurement uncertainties can be found in the supplement.

### 2.5 Chemical composition

Aerosol particle dry deposition was collection with a flat plate type sampler (Ott and Peters, 2008) on carbon adhesive mounted to standard electron microscopy stubs. Sample substrates were exposed for approximately 24 hours, collecting particles approximately between 1 and 100 $\mu$m particle diameter at ambient conditions. Samples were subject to automated electron microscopy single particle analysis, yielding particles size (projected area diameter) and average elemental composition for each particle. Particles were classified according to the composition in group based on a static rules set. For more information on sampling, analysis and data processing refer to Kandler et al. (2018). In this study, we calculated the particle mass deposition rate in the size range from 1 to 8 $\mu$m.

## 3 Results and discussion

### 3.1 Overview of the meteorology and air quality

Time series of meteorological and air quality parameters as measured from 2 to 30 April are shown in Fig. 2. The relative humidity (RH), temperature, wind speed, wind direction, $NO_x$ and $N_{\mathrm{total}}$ (retrieved from MPSS and APS measured PNSD) were determined at the measurement site. Note that all times presented here are in UTC (corresponding to local time$-3$).

RH exhibited large variability throughout the campaign, varying from 22.6% to 89.2%, with a mean of 68.4%. Temperature varied from 10.0 to 26.5 °C, with a mean of 17.5 °C. The local wind speeds ranged from 0.1 to 10.1 m s$^{-1}$, with a mean of 2.8 m s$^{-1}$. Fig. S3 shows the wind rose plot based on 10 minutes mean of wind speed and wind direction. It is clear that winds



are mainly from northwest, west and northeast. The winds from northwest and west featured higher wind speeds while winds from northeast featured lower wind speeds.

$NO_x$ varied from 0.0027 to 25 ppbv, with a median of 0.67 ppbv. $N_{total}$ varied from 658 to 61308 $cm^{-3}$, with a median of 3954 $cm^{-3}$. The $NO_x$ and $N_{total}$ were relatively low during most of the campaign. However, sharp increases in $NO_x$ and $N_{total}$

were observed frequently and extremely high concentrations ($NO_x$>1.6 ppbv, $N_{total}$>8000 $cm^{-3}$) only occurred for few hours. A good correlation ($R^2$=0.62) was found between such extremely high concentrations of $NO_x$ and $N_{total}$ (Fig. S4), indicating a nearby pollution source. The extremely high concentrations of $NO_x$ and $N_{total}$ together with the wind direction typically connected to their occurrences, suggests the nearby airport as the source for these pollutions, as will be discussed in more detail in Sec. 3.2.

To get indications concerning possible particle sources, we studied the air mass origin and transport by means of backward trajectory analysis. The calculations were performed with the HYSPLIT (HYbrid Single-Particle Lagrangian Integrated Trajectory) Model (Stein et al., 2015; Rolph, 2003). Fig. 3(a) shows the 6-day backward trajectories with 1 hour time resolution ending at 500 m above the measurement site. Fig. 3(b) shows the relative frequency of backward trajectories. The majority (more than 30%) of the trajectories featured paths over central and southern Europe. Around 10% of the trajectories were

traced back to the northern Atlantic Ocean and travelled through the western Mediterranean Sea to the site. Less than 5% of the trajectories touched the Sahara Desert and the desert regions in Syria and Iraq, indicating that mineral dust particles could have been transported to Cyprus during the campaign.

## 3.2 Particle number size distribution and sources

Particles of different sizes have different formation pathways, sources and behaviors. Fig. 4(a) presents measured super-micron

PNSDs as contour plot, together with wind speed information. The super-micron particle concentration varied from 0 to 11 $cm^{-3}$, with a mean of 2 $cm^{-3}$. Fig. 5 shows the time series of particle mass deposition rate for different compounds at Cyprus, for particles between 1 and 8 $\mu$m dry diameter. Overall, sea salt accounted for more than 60% of the super-micron particle mass throughout the whole campaign.

Higher super-micron particle number concentrations were mainly observed from 6 to 7, 12 to 14 and 21 to 22 April, with

the corresponding air masses originating from the Sahara Desert or the desert regions in Syria and Iraq, as shown in Fig. 4(a) by brown dots. As shown in Fig. 5, high dust deposition rates of $\sim$1 mg $m^{-2}$ $d^{-1}$ were also observed during these periods. Therefore, mineral dust was another important constituent of super-micron particle mass during these periods. However, the observed super-micron particle concentrations were relatively low compared to those reported in previous studies (Mamouri et al., 2016; Schrod et al., 2017) for Cyprus during dust plumes. Low concentrations of super-micron particles were observed

on 15 April although the respective backward trajectories featured paths over the Sahara dust region. In summary, the super-micron particles observed during the campaign, were a mixture of $\sim$60% sea salt, $\sim$32% mineral dust and $\sim$8% others (mainly sodium sulfate), with the relative contributions being dependent on the actual meteorological conditions and source regions.

Fig. 4(b) presents contour plots of PNSDs observed for submicron particles. Extremely high concentrations of ultrafine particles (pronounced mode with a maximum at about 15 nm, median dN/dlogDp value larger than $10^4$ $cm^{-3}$), were frequently



observed throughout the whole campaign. When ultrafine particles featured high concentrations, extremely high concentrations of $NO_x$ were also observed. An exemplary case is shown in Fig. S5. Such kind of behavior usually appeared from 03:00 to 06:00 UTC and 17:00 to 22:00 UTC. A wind rose plot shown in the supplement indicates that during these periods, winds were from the northeast (Fig. S6), i.e., the direction where the Paphos International Airport is located. This is highly suggestive for

the airport being the origin of the observed ultrafine particles and $NO_x$. Fig. 6 shows the comparison of medians of PNSDs observed during airport affected (PNSDa) and non-affected time periods. The error bars indicate the range between the 25% and 75% percentiles. It is clearly seen that airport affected PNSDa exhibit a very pronounced ultrafine particle mode with a maximum at diameters of about 15 nm. Such a mode is indicative for a nearby particles source, such as the combustion of fuel at the airport. Previous studies found that airport emitted particles featured similar PNSDs (Hudda and Fruin, 2016; Jasinski

and Przylebska, 2018). Therefore, in the following, time periods affected by pollution from the airport were excluded from further analysis. The pollution-free median PNSD (black line in Fig. 6) features clear Atiken, accumulation and coarse modes, with the Hoppel minimum (Hoppel et al., 1986) being located at approximately 80 nm.

Based on the criteria reported by Dal Maso et al. (2005), we identified several NPF and growth events during the campaign. The criteria are, first of all, the appearance of a distinct new mode (in the nucleation mode size range) in the size distribution.

Secondly, the mode must prevail over a time span of hours. Lastly, the new mode must show signs of growth. For example, newly formed particles occurred at 07:00 UTC 5 April, 08:00 UTC 12 April and 07:00 UTC 22 April, with subsequent particle growth in the next few hours up to days. All observed NPF started during daytime, suggesting that photochemistry products were likely to contribute to the formation of the new particles. The NPF events, which occurred at 07:00 UTC 5 April and 07:00 UTC 22 April, featured continuous particle growth up to several tens of nanometers. The NPF event occurring at 08:00 UTC

12 April exhibits a more complicated time evolution. Around 15:30 UCT 12 April, the PNSDs were affected by pollution from the airport due to the wind direction shifting to the northeast. Around 00:00 UTC 13 April, the wind speed increased and wind directions were from the clean ocean, i.e., clean air mass weakened the particle growth process. Later on, i.e., at 01:00 UTC 14 April, precipitation occurred. This influenced the evolution of the NPF and growth event, but the growing trend in particle size is still to be seen. The observed particle growth events show that newly formed particles can grow up to sizes where they can

act as CCN. However, there are several more NPF and growth events which we do not discuss here, because particles did not grow up to sizes making them potential CCN.

### 3.3  CCN and particle hygroscopicity

Fig. 7 shows time series of $N_{total}$ and $N_{CCN}$ (corrected with particle losses) in the upper panel, $d_{crit}$ in the middle panel, and $\kappa$ in the lower panel. $N_{CCN}$ exhibit large variability throughout the campaign, including a few remarkably elevated concentrations

(maximum value $\sim$3730 cm$^{-3}$ at supersaturation of 0.31%), and one exceptionally low concentration (minimum value $\sim$170 cm$^{-3}$ at supersaturation of 0.31%). The median values of $N_{CCN}$ at different supersaturation are given in Tab. 1, and vary from 295 cm$^{-3}$ for a supersaturation of 0.08% to 2004 cm$^{-3}$ for a supersaturation of 0.77%.

The low $N_{CCN}$ around 03:00 UTC 14 April was associated with precipitation as can be seen in Fig. 2. Most of the time, high $N_{CCN}$ are associated with NPF and growth events. For example, around 09:00 UTC 5 April, $N_{CCN}$ at higher supersaturations





(0.54% and 0.77%) started to increase. The $N_{CCN}$ at lower supersaturations (0.19% and 0.31%) followed at 04:00 UTC 6 April. However, $N_{CCN}$ at the lowest supersaturation (corresponding the $d_{crit}$ around 163 nm) did not increase in connection with the NPF and growth event. Newly formed particles did not grow into that size range, i.e., $N_{CCN}$ at the lowest supersaturation was not affected. The same behavior was observed from 08:00 UTC 22 to 00:00 UTC 23 April. From 13 to 14 April, the NPF and

growth were affected by changing wind directions and precipitation. $N_{CCN}$ also shows respective influences, but the overall trend still can be seen.

The probability density functions (PDFs) of $N_{CCN}$ at different supersaturations are shown in the upper panel of Fig. 8. As discussed, $N_{CCN}$ at lowest supersaturation was not affected by the NPF and growth events, so a unimodal PDF was observed. However, the PDFs of $N_{CCN}$ at other supersaturations are bimodal, with the larger mode (higher concentrations) representing

the NPF and growth events. Kalivitis et al. (2015) also found that CCN production resulted from NPF in the eastern Mediterranean during the summertime. The small mode (lower concentrations) of the PDFs are representative for the time periods without NPF and growth events.

The $d_{crit}$ at different supersaturations were almost constant throughout the campaign, even during the NPF events. The PDFs of $d_{crit}$ are unimodal, as shown in Fig. 8. The $d_{crit}$ at different supersaturations, and the standard deviations of their PDFs, are

included in Tab. 1. For the supersaturations of 0.77% and 0.54%, the $d_{crit}$ were below 60 nm, i.e., inside the Aitken mode. However, for the lowest supersaturation of 0.08%, $d_{crit}$ is located in the accumulation mode. Consequently, hygroscopicities derived at these supersaturations, can be assumed to be representative for the Aitken (at supersaturations of 0.77% and 0.54%) and the accumulation mode (at a supersaturation of 0.08%), respectively.

The particle hygroscopicity, expressed as $\kappa$, can be seen as a measure for average particle chemical composition. Time series

of calculated $\kappa$ values are depicted in the lower panel of Fig. 7. The $\kappa$ values at different supersaturations show little variability over time, with 1 standard deviation from 0.09 to 0.13, i.e., there is no clear trend in $\kappa$ over time during the campaign. At the supersaturations of 0.54% and 0.77%, corresponding to $d_{crit}$ of 40±8 and 55±7 nm (median±1 standard deviation), the medians of $\kappa$ are 0.21±0.10 and 0.29±0.10, respectively. These low $\kappa$ values in Aitken mode suggest the presence of organic material, which has also been observed in previous studies (Kalivitis et al., 2015; Kristensen et al., 2016). At the lowest

supersaturation of 0.08%, corresponding to the $d_{crit}$ of 163±10 nm, the median of $\kappa$ is 0.57±0.09. Particles in this size range are members of the accumulation mode, and have undergone cloud processing and aging. This results in higher amounts of sulfates being present, and consequently higher hygroscopicities. A clear downward trend of $\kappa$ is observed with increasing supersaturations and decreasing $d_{crit}$ (Fig. 9). The $\kappa$ values in the Aitken and accumulation modes are clearly different, with the error bars considered, indicating significant differences in particle chemical composition for the two modes.

The PDFs of $\kappa$ change from unimodal to bimodal to unimodal with decreasing supersaturation. As mentioned above, the $\kappa$ values at supersaturations of 0.77% and 0.54% are representative for the Aitken mode particles, while the $\kappa$ values at supersaturation of 0.08% are a measure for the accumulation mode particles. Therefore, the $\kappa$ values at these supersaturations feature unimodal distributions. $\kappa$ at supersaturations of 0.31% and 0.19%, corresponding to $d_{crit}$ of 92±8 and 70±8 nm, respectively, exhibit bimodal distributions. These $\kappa$ values are influenced by both Aitken and accumulation mode particles,

indicating an external mixture of particles in that size range.





The determined particle hygroscopicities confirm those given in previous studies. For example, Kalivitis et al. (2015) reported that $\kappa$ values in the Aitken mode were 0.20-0.40 lower than those in the accumulation mode during the NPF events in the eastern Mediterranean, and highlighted NPF as a source of CCN. Pringle et al. (2010) used an atmospheric chemistry model to derive global distributions of effective particle hygroscopicity $\kappa$. The annual mean value at the surface of the eastern

Mediterranean was roughly 0.45, with an annual cycle ranging from 0.35 in December to 0.50 in February. For April, the period of this study, the value of 0.40 was reported, which is consistent with what we obtained ($\kappa$=0.39) for this campaign.

### 3.4 Ice nucleating particles

#### 3.4.1 Temperature spectra of cumulative $N_{INP}$

Ice fractions ($f_{ice}$) as determined with both, LINA and INSEKT, are shown in Fig. S7. The corresponding $N_{INP}$ from both

instruments are shown in Fig. 10 as a function of temperature. Samples collected from the land and ocean sectors (measured by LINA) are represented by black circles and red rectangles, respectively. These filter samples were all active at $-16\,°C$ and the highest freezing temperature was found to be $-6.5\,°C$. Samples collected during day time (measured by INSEKT) are represented by blue rectangles. With two or three dilution steps, by measuring suspensions with different aerosol concentrations, the INSEKT measurements cover a larger temperature range, from $-7.5\,°C$ to $-26.5\,°C$. The measurement uncertainty for

both instruments is shown in Fig. S8. As mentioned in the experimental section, filters examined with LINA were switched according to wind direction. From Fig. 10, it is obvious that there is no very pronounced difference in $N_{INP}$ between the land and ocean sectors. It is, however, noticeable that the freezing curves from the ocean sector are rather at the lower end of the measured curves. To test if there was a pronounced contribution to INP from the land sector, we examined the INSEKT data in more depth. Fig. S9 shows the $N_{INP}$ from the INSEKT measurements in dependence on the fraction of time sampled from

the ocean sector. No clear trend was found. These observations are indicative for the absence of nearby sources, and hence we conclude that the sampled INP, at least those ice active at $< -15\,°C$, originate from long-range transport.

The measured $N_{INP}$ in this study are within the $N_{INP}$ range presented by Welti et al. (2018), who characterized INP sampled at the Cape Verde Atmospheric Observatory (CVAO) over a time period of 4 years (shown in Fig. 10 as yellow shadow). This is surprising as those atmospheric aerosols at CVAO and Cyprus should be expected to be different. It might, however, point

towards a similar background of INP worldwide. $N_{INP}$ are lower than those proposed in Fletcher (1962), while the slope is similar to that of the Fletcher (1962) line. $N_{INP}$ increased exponentially from $-10$ to $-25\,°C$, indicating the presence a of broad variety of INP, featuring e.g., different size, composition, ice active surface sites.

$N_{INP}$ at a particular temperature span about 1 order of magnitude below $-20\,°C$, and about 2 orders of magnitude at the warmer temperatures (T$> -18\,°C$). This is consistent with the previous study of O' Sullivan et al. (2018), who carried out

field measurement in northwestern Europe. Few samples (LINA sample05, 20, 22 and INSEKT sample01, 06, 12, 13, 19, 28) showed elevated concentrations at temperatures above $-15\,°C$. Biological particles (e.g., bacteria, fungal spores, pollen, viruses, and plant fragments) usually contributed to the INP at this moderate supercooling temperatures (Kanji et al., 2017; O' Sullivan et al., 2018). These high signals observed in both instruments might have been caused by biogenic INP, originating





from the Cyprus island, as such high signals did not occur for the four samples from the ocean sector. However, as there are only four samples from the ocean sector, and as no additional tests were possible with the limited amount of sampled material, it should suffice to express this hypothesis here.

Overall, $N_{\mathrm{INP}}$ of the land samples are not clearly different from those of the ocean samples, besides for some samples at
> −15 °C for which a biogenic contribution is expected. Therefore, a contribution of INP from pollution from the airport is not expected. This would be in line with Chen et al. (2018), who found that aerosol in Beijing did not contain higher $N_{\mathrm{INP}}$ during strong pollution events, compared with clean phases.

### 3.4.2  Time series and PDFs

Fig. 11(a) shows the time series of $N_{\mathrm{INP}}$ during the campaign. Here we present $N_{\mathrm{INP}}$ derived from LINA (ocean sector in green and land sector in red) and INSEKT (in blue) measurements at −15, −18 and −20 °C. $N_{\mathrm{INP}}$ varied from 0.001 to 0.1, 0.004 to 0.2 and 0.03 to 0.4 std L$^{-1}$ at −15, −18 and −20 °C, respectively. $N_{\mathrm{INP}}$ varies non-synchronously at different temperatures. Here we compared data from different temperatures with each other and determine a regression line between them. Taking, e.g., the results from the LINA measurements, the coefficient of determination (R$^2$) are 0.45, 0.26 and 0.0033 for −15 to −18 °C, −18 to −20 °C, −15 to −20 °C, indicating the different natures and origins of the INP active at different temperatures.

Welti et al. (2018) found that log-normal distributions best approximate the measured variability in concentrations at each individual temperature. Here we used two methods to test our $N_{\mathrm{INP}}$ frequency distributions, which are both described in more detail in the supplemental information. Both methods indicate that the INP distributions at −15, −18 and −20 °C are indeed log-normally distributed. This analysis was only done for these temperatures, as only in this temperature range, almost all samples contributed data. As log-normally distributed $N_{\mathrm{INP}}$ are indicative for the observed INP population having undergone a series of random dilutions while being transported (Welti et al., 2018), the performed tests yield prove for the INP (ice active at −20 ≤T≤−15 °C) sampled during our measurements originating from long-range transport rather than local sources.

Fig. 11(b) depicts the PDFs of $N_{\mathrm{INP}}$ at different temperatures. Thereby, a PDF is shown, if at the particular temperature, most of investigated samples featured a quantifiable (0<$f_{\mathrm{ice}}$<1) freezing behavior. For example, there were three LINA-measured samples which did not freeze at −15 °C ($f_{\mathrm{ice}}$=0), therefore, we do not show the PDF of LINA-measured $N_{\mathrm{INP}}$ at −15 °C. At −20 °C the data from Welti et al. (2018) is omitted, because more than half of all samples were fully frozen ($f_{\mathrm{ice}}$=1). As can be seen from Fig. 11(b), our results are comparable to those given in Welti et al. (2018) (black curves) derived from long-term measurement at CVAO. Note that it is not possible to directly compare the $N_{\mathrm{INP}}$ measured by LINA and INSEKT, as they always had different sampling times and INSEKT always sampled air from all directions whereas LINA got it from the different sectors separately. But in general, no systematic deviation can be seen, as can be seen when looking at the PDFs. To the best of our knowledge, the only in-situ observations at −20 °C for supersaturated conditions (101%) in the eastern Mediterranean was reported by Schrod et al. (2017) during a heavy dust plume at high altitude with 0.03 to 3 std L$^{-1}$.





### 3.4.3 Correlation of $N_{\mathrm{INP}}$ with particle number/surface area concentration and parameterization

Scatter plots of LINA- and INSEKT-measured $N_{\mathrm{INP}}$ at temperatures of $-15$, $-18$ and $-20\,^\circ\mathrm{C}$ against particle number concentration in the size range $>500$ nm ($N_{>500\mathrm{nm}}$) are shown in Fig. 12(a) and Fig. 12(b). The averaged $N_{>500\mathrm{nm}}$ during each filter sample varied from 2 to 14 cm$^{-3}$. The $N_{>500\mathrm{nm}}$ in this study is much lower than that observed during the dust plume period in

Cyprus (maximum 75 cm$^{-3}$ Schrod et al., 2017). The R$^2$ between $N_{>500\mathrm{nm}}$ and $N_{\mathrm{INP}}$ are shown in Tab. S4. The R$^2$ were all below 0.25, indicating no correlation between $N_{\mathrm{INP}}$ and $N_{>500\mathrm{nm}}$.

Based on nine field studies occurring at a variety of locations over 14 years, DeMott et al. (2010) proposed a parameterization of the "global" average INP distribution. Besides, Tobo et al. (2013) present a similar parameterization method with adjusted coefficients to predict INP populations in a forest ecosystem. Fig. 12(c) and Fig. 12(d) compare the $N_{\mathrm{INP}}$ we measured with

LINA and INSEKT to the predicted $N_{\mathrm{INP}}$ on the basis of the DeMott et al. (2010) and Tobo et al. (2013) parameterizations. As can be seen, the DeMott et al. (2010) parameterization overestimates the observed values by about 2 orders of magnitude on average. The Tobo et al. (2013) parameterization can reproduce only 24% and 25% of the $N_{\mathrm{INP}}$ measured by LINA and INSEKT within a factor of 2, respectively. The Tobo et al. (2013) parameterization overestimates the observed values about 1 order of magnitude on average. This, together with $N_{\mathrm{INP}}$ not being correlated with $N_{>500\mathrm{nm}}$ (see Tab. S4), indicates that the application

of parameterizations in connection with measured particle number concentrations has to be done with extreme caution, as the encountered particle populations may significantly differ from those considered when developing the parameterizations.

Fig. S11 shows the median particle surface area size distribution (PSSD) for the whole campaign (excluding the airport pollution events). Two different modes were observed, i.e., a small mode (20-500 nm) and a larger mode (500-7000 nm). Based on the PSSD, the concentrations for the total surface area of the small mode ($S_{<500\mathrm{nm}}$), the large mode ($S_{>500\mathrm{nm}}$) and

for both modes combined ($S_{\mathrm{all}}$) were calculated. The $S_{<500\mathrm{nm}}$ is about 4 times higher than $S_{>500\mathrm{nm}}$. Scatter plots of LINA and INSEKT measured $N_{\mathrm{INP}}$ against $S_{<500\mathrm{nm}}$, $S_{>500\mathrm{nm}}$ and $S_{\mathrm{all}}$ are shown in Fig. S12(a) and Fig. S12(b). The R$^2$ between $N_{\mathrm{INP}}$ and particle surface area concentration are shown in Tab. S5. The R$^2$ are all below 0.20, indicating no correlation between $N_{\mathrm{INP}}$ and particle surface area concentration.

The ice nucleating properties of aerosol particles may be characterized by its ice active surface site density ($n_{\mathrm{s}}$). The $n_{\mathrm{s}}$ is a

measure of how well an aerosol acts as a seed surface for ice nucleation. The $n_{\mathrm{s}}$ can be calculated as:

$$n_{\mathrm{s}} = \frac{N_{\mathrm{INP}}(\theta)}{S} \qquad (4)$$

Where $S$ is the particle surface area concentration.

Depending on which particle size range was investigated, previous studies calculated $n_{\mathrm{s}}$ based on either the total surface area concentration ($S_{\mathrm{all}}$) or on the surface area concentration of particles larger than 500 nm ($S_{>500\mathrm{nm}}$). Here, both approaches were

used, resulting in $n_{\mathrm{s\_all}}$ and $n_{\mathrm{s>500\,nm}}$, respectively. Fig. 13 shows the $n_{\mathrm{s>500\,nm}}$ as black box plot and the $n_{\mathrm{s\_all}}$ as red box plot at $-15$, $-18$ and $-20\,^\circ\mathrm{C}$. As can be seen, $n_{\mathrm{s}}$ increases towards lower temperature, which is expected. The $n_{\mathrm{s}}$ results, calculated using LINA and INSEKT measured $N_{\mathrm{INP}}$, are shown in Fig. 13(a) and Fig. 13(b), respectively. The $n_{\mathrm{s}}$ values determined from LINA measurements are consistent with those from INSEKT measurements.



To the best of our knowledge, many studies dealt with the $n_s$ for dust aerosol particles, while no study investigated the $n_s$ for anthropogenically polluted aerosol. In the following, we compare our $n_{s\_all}$ for the anthropogenically polluted aerosol on Cyprus, with $n_{s\_all}$ based on existing parameterizations (Niemand et al., 2012; Ullrich et al., 2017) for dust aerosols (Fig. 13). However, the $n_{s\_all}$ values from the parameterizations are more than 2 orders of magnitude larger than the $n_{s\_all}$ found in this study. Price et al. (2018) carried out an airborne measurement in dust laden air over the tropical Atlantic. The $n_{s\_all}$ reported in Price et al. (2018) (shown in Fig.13 as yellow shadow) is about 1 to 2 orders of magnitude higher than our results. Based on airborne measurement, Schrod et al. (2017) found that the $n_{s>500nm}$ at Cyprus ranged between $10^5$ to $10^8$ m$^{-2}$ at T=$-20$ °C, RH$_{water}$ =101 %, shown as green shadow in Fig. 13.

In short summary, parameterizations purely based on $N_{>500\,nm}$ or particle surface area concentration in mineral dust dominated model systems overestimate the $N_{INP}$ in the anthropogenically polluted aerosol on Cyprus. This was also found in a different context, anthropogenically polluted air masses in Being (Chen et al., 2018), and is based on the fact that more strongly anthropogenically influenced air masses have larger numbers of particles in the size range above 500 nm than naturally ones.

## 4    Conclusions

The A-LIFE campaign took place in April 2017 on the island of Cyprus to investigate the aerosols prevailing in the eastern Mediterranean region. As part of the A-LIFE campaign, ground-based measurements were carried out in Paphos, Cyprus, to characterize the abundance, properties (size distribution, hygroscopicity, ice activity), and sources of aerosol particles in general, CCN and INP in particular.

During these activities, frequently NPF and growth events were observed. Following NPF, during some events, on time scales of few hours to days, particles grew into the CCN size range. In fact, the highest observed $N_{CCN}$ were connected with NPF and growth events, which confirms the importance of NPF as source of CCN in the eastern Mediterranean.

Usually, trimodal (Aitken, accumulation, coarse mode) PNSDs were observed. Aitken mode particles featured low hygroscopicities ($\kappa$ values about 0.21 to 0.29), indicating the presence of organic materials. Accumulation mode particles featured higher $\kappa$ values of about 0.57, indicating that particles in the accumulation mode underwent cloud processing and aging, resulting in higher amounts of sulfate being present. The super-micron particles were mainly from SSA and partly mineral dust.

PDFs of $\kappa$ in both, the Aitken and the accumulation mode, exhibit a unimodal structure, while the $\kappa$-PDFs for particles sizes close to the Hoppel minimum, feature a bimodal shape. This indicates the presence of both, non-cloud-processed (Aitken mode) and cloud-processed (accumulation mode particles), in the size range around the Hoppel minimum. The average observed $\kappa$ of 0.39 confirms values found in previous field measurements (Kalivitis et al., 2015) and in model results (Pringle et al., 2010) for the Mediterranean region.

Atmospheric $N_{INP}$ where determined in the temperature range from $-6.5$ to $-26.5$ °C, using two freezing array type instruments (LINA, TROPOS, and INSEKT, KIT). $N_{INP}$ at a particular temperature span around 1 order of magnitude below $-20$ °C, and about 2 orders of magnitude at warmer temperature (T$> -18$ °C). Few samples showed elevated concentrations



at temperatures T$> -15$ °C, which suggests a significant contribution of biological particles to the INP population, which might have originated from the Cyprus island. No significant difference in $N_{\mathrm{INP}}$ were found when selectively sampling wind directions from the land or sea sector for INP that were ice active in the temperature range between $-15$ and $-20$ °C. PDFs of $N_{\mathrm{INP}}$ at a particular temperature follow log-normal distributions. For example, of at $-18$ °C, the $N_{\mathrm{INP}}$ ranged from 0.004 to

0.2 std L$^{-1}$ during the campaign, which is consistent with the previous study of Welti et al. (2018). This indicates, that these sampled INP which are ice active below $-15$ °C originate from long-range transport rather than local sources.

No correlations were found between $N_{\mathrm{INP}}$ and $N_{>500\mathrm{nm}}$. Parameterizations (DeMott et al., 2010; Tobo et al., 2013) based on $N_{>500\mathrm{nm}}$ were found to overestimate the $N_{\mathrm{INP}}$ by about 1 to 2 orders of magnitude. There was also no correlation between $N_{\mathrm{INP}}$ and particle surface area concentration. The $n_{\mathrm{s}}$ for anthropogenically influenced aerosol on Cyprus was found to be 1

to 3 orders of magnitude lower than the $n_{\mathrm{s}}$ for dust aerosol particles resulting from previous studies (Niemand et al., 2012; Ullrich et al., 2017; Price et al., 2018). This clearly highlights, that usage of such parameterizations, just based on measured particle number or surface area size distributions, is not always feasible for predicting $N_{\mathrm{INP}}$, as the parameterizations where derived for particular aerosol types. In other words, basing modelling efforts on, e.g., PDFs from observed $N_{\mathrm{INP}}$, rather than on parameterizations, might be the method of choice, if the aerosol particle and/or INP composition is unknown.

*Data availability.*   The data are available through the World Data Center PANGAEA (https://www.pangaea.de/) in the near future. A link to the data can be found under this paper's assets tab on ACP's journal website.

*Author contributions.*   X. Gong wrote the manuscript with contributions from T. Müller, A. Wiedensohler, K. Höhler, K. Kandler, H. Wex and F.Stratmann. N. Ma and T. Müller performed particle number size distribution measurement and X. Gong performed data evaluation. Chemical composition measurements and data evaluation were performed by K. Kandler. CCN measurements and data analysis were per-

formed by X. Gong. LINA measurements and data evaluation were performed by X. Gong. INSEKT measurements and data evaluation were performed by B. Dietel, T. Schiebel, K. Höhler and X. Gong. X. Gong, H. Wex and F. Stratmann discussed the results and further analysis after the campaign. All co-authors proofread and commented the manuscript.

*Competing interests.*   The authors declare that they have no conflict of interests.

*Acknowledgements.*   The works were carried out in the framework of the A-LIFE project. This project has received funding from the Euro-

pean Research Council (ERC) under the European Union's Horizon 2020 research and innovation programme under grant agreement No. 640458. We would like to thank Dr. Umar Javed from Institute of Energy and Climate Research, Troposphere (IEK-8) and Dr. Anywhere Tsokankunku from Max Plank Institute for Chemistry to provide the NO$_{\mathrm{x}}$ data. K. Kandler is funded by the Deutsche Forschungsgemeinschaft (DFG, German Research Foundation) – 264907654; 264912134; 378741973; 416816480.





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





**Table 1.** Median values of $N_{\mathrm{CCN}}$, $d_{\mathrm{crit}}$, $\kappa$ and one standard deviation of $d_{\mathrm{crit}}$ and $\kappa$ at different supersaturations.

| Supersaturation (%) | $N_{\mathrm{CCN}}$ (cm$^{-3}$) | $d_{\mathrm{crit}}$ (nm) | $\kappa$ | $\sigma_{d_{\mathrm{crit}}}$ (nm) | $\sigma_\kappa$ |
|---|---|---|---|---|---|
| 0.08 | 295 | 163 | 0.57 | 10 | 0.09 |
| 0.19 | 872 | 92 | 0.49 | 8 | 0.12 |
| 0.31 | 1332 | 70 | 0.42 | 8 | 0.13 |
| 0.54 | 1743 | 55 | 0.29 | 7 | 0.10 |
| 0.77 | 2004 | 48 | 0.21 | 8 | 0.10 |





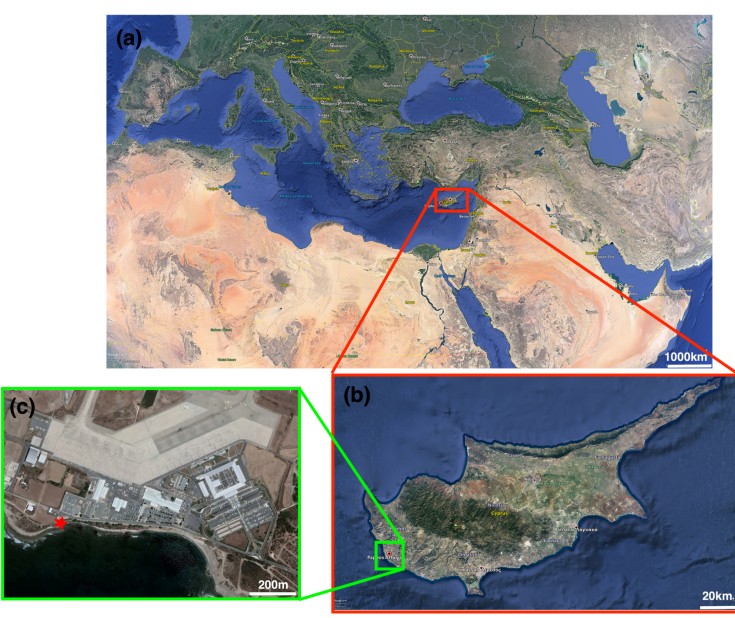

**Figure 1.** Maps of the Mediterranean region, Cyprus and sampling location. (a) Position of Cyprus in the Mediterranean region. (b) Position of Paphos city in the Cyprus island. (c) The sampling site is displayed as red star. On the northeast of the sampling site is the Paphos International Airport.





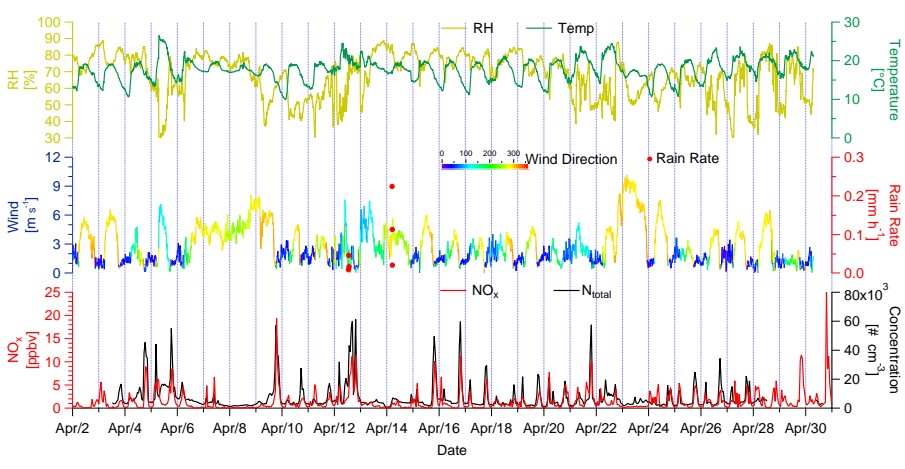

**Figure 2.** Time series of RH, temperature, wind speed, and wind direction with 10 minutes resolution, NO$_x$ and $N_{total}$ with 1 hour resolution,
.





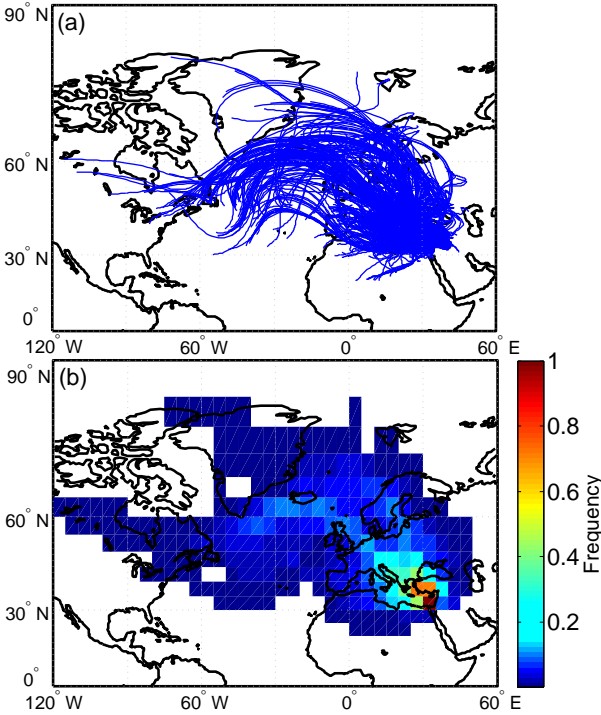

**Figure 3.** (a) 6-day backward trajectories (blue lines) ended at 500 m above the measurement station with 1 hour resolution. (b) Relative frequency of trajectories arriving at the station, based on a 5° by 5° grid size.



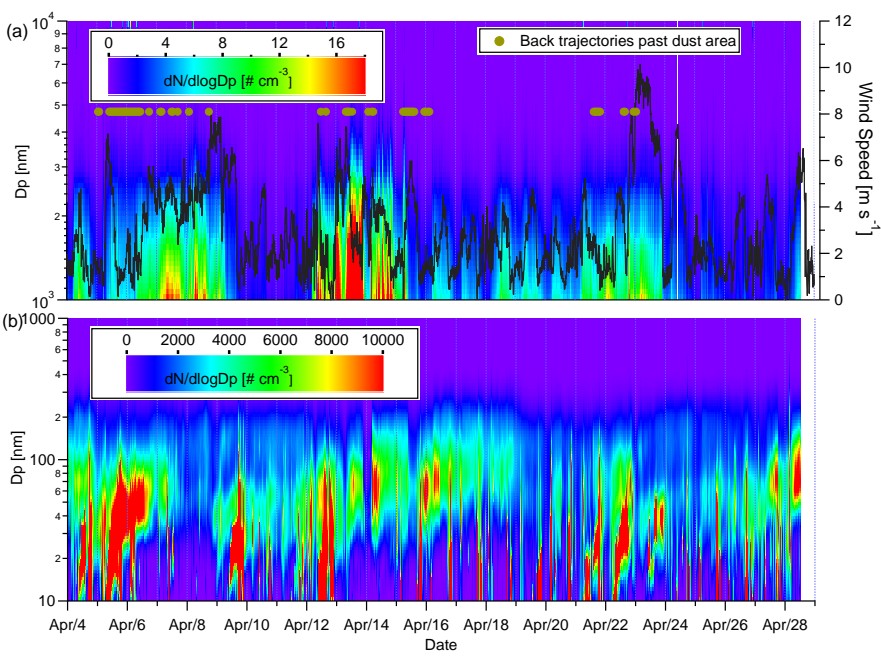

**Figure 4.** Contour plots for PNSDs during the whole campaign. The color scale indicates dN/dlogDp in cm$^{-3}$. (a) Contour plots for PNSDs of 1000 to 10000 nm. Black line shows time series of wind speed and the brown dots show the time when backward trajectories passed the dust area. (b) Contour plots for PNSDs of 10 to 1000 nm.





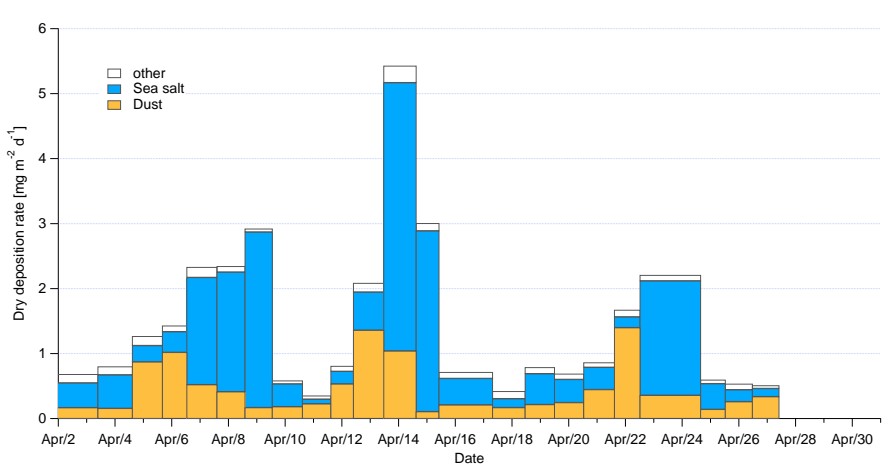

**Figure 5.** Time series of dry mass deposition rate for different compounds for particles between 1 and 8 $\mu$m dry diameter. The 'Dust' class includes silicate and carbonate particles, the 'Other' class mainly consists of sodium sulfate. Mixed particles are evenly distributed between the according groups.



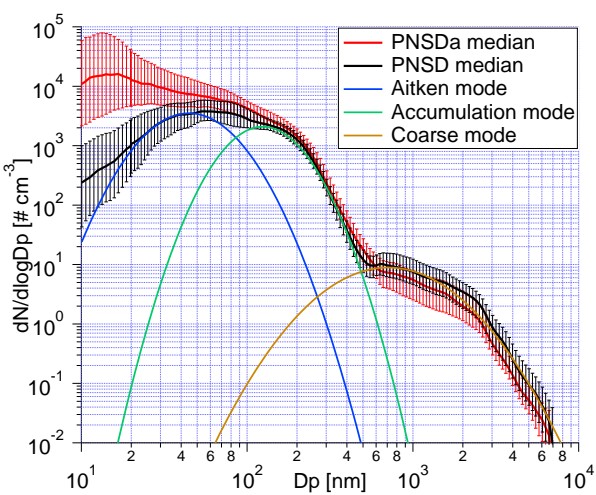

**Figure 6.** Comparison of the median PNSD during airport affected (red line) and non-affected (black line) time periods . The error bar indicates the range between the 25 % and 75 % percentiles. Aitken, accumulation and coarse modes are fitted with log-normal distribution, displayed in blue, green and brown lines, respectively.



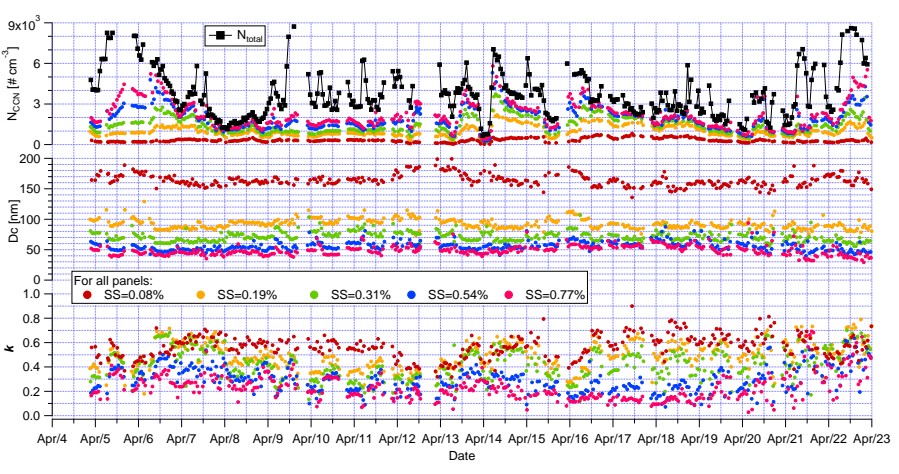

**Figure 7.** Time series of $N_{\text{total}}$, $N_{\text{CCN}}$, the inferred $d_{\text{crit}}$ and $\kappa$ values at different supersaturations.





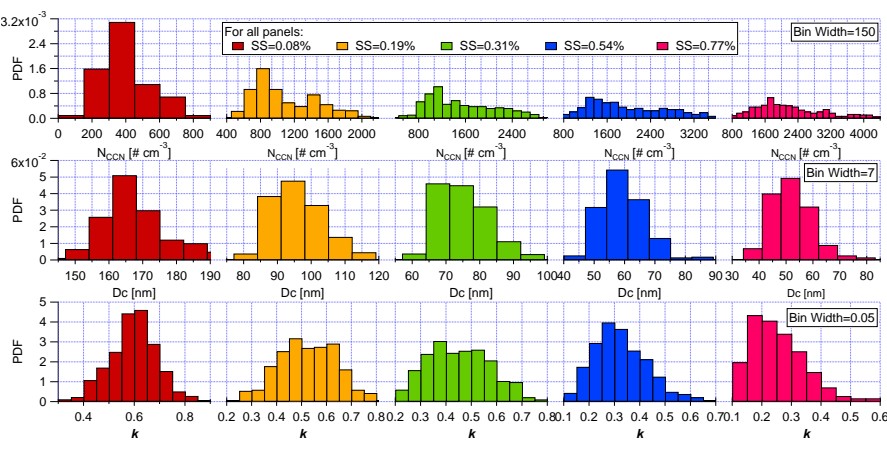

**Figure 8.** PDFs of $N_{CCN}$, $d_{crit}$ and $\kappa$ values at different supersaturations.





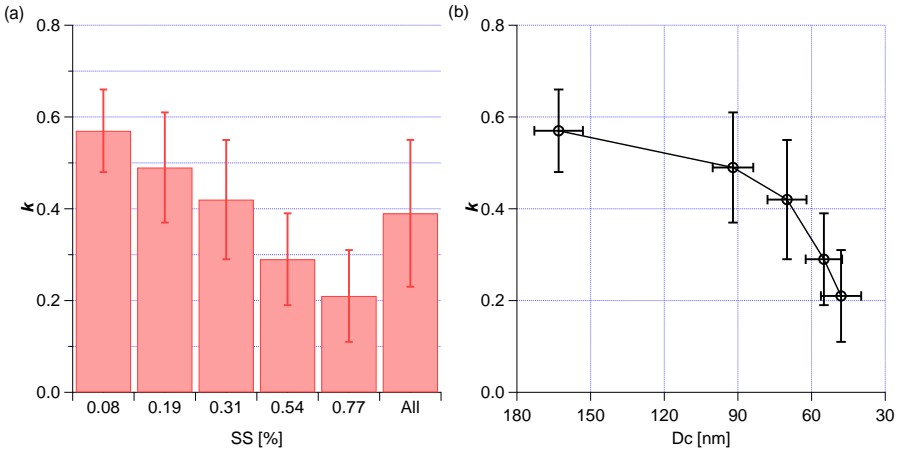

**Figure 9.** (a) Derived $\kappa$ values at different supersaturations. (b) $\kappa$ values as a function of corresponding $d_{\mathrm{crit}}$. Error bar represents the one standard deviation.





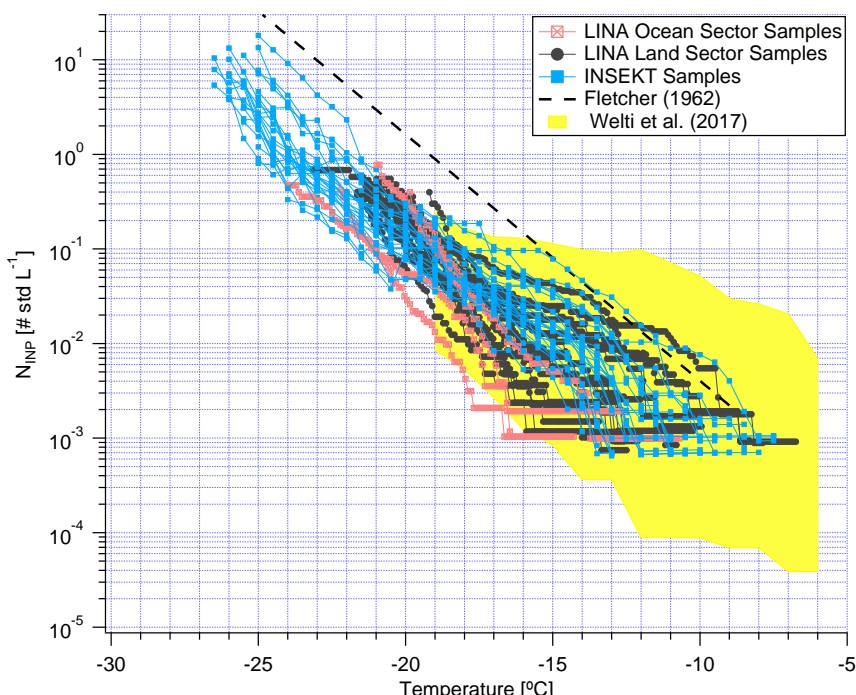

**Figure 10.** $N_{INP}$ (measured by LINA and INSEKT) as a function of temperature. Parameterization from Fletcher (1962) in the valid temperature range is given for comparison, as shown in dashed line. The yellow shadow represents the measured $N_{INP}$ from a ground-based station at CVAO (Welti et al., 2018).



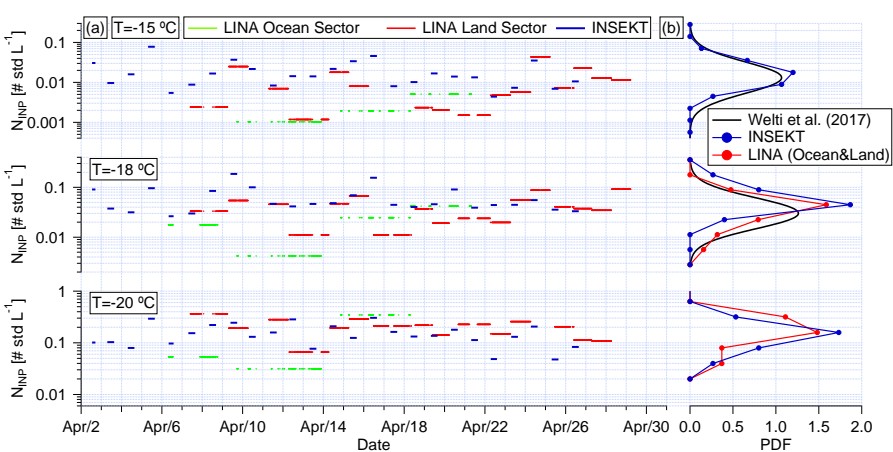

**Figure 11.** (a) Time series and (b) PDF of $N_{\mathrm{INP}}$ at $-15$, $-18$ and $-20\,^\circ$C.





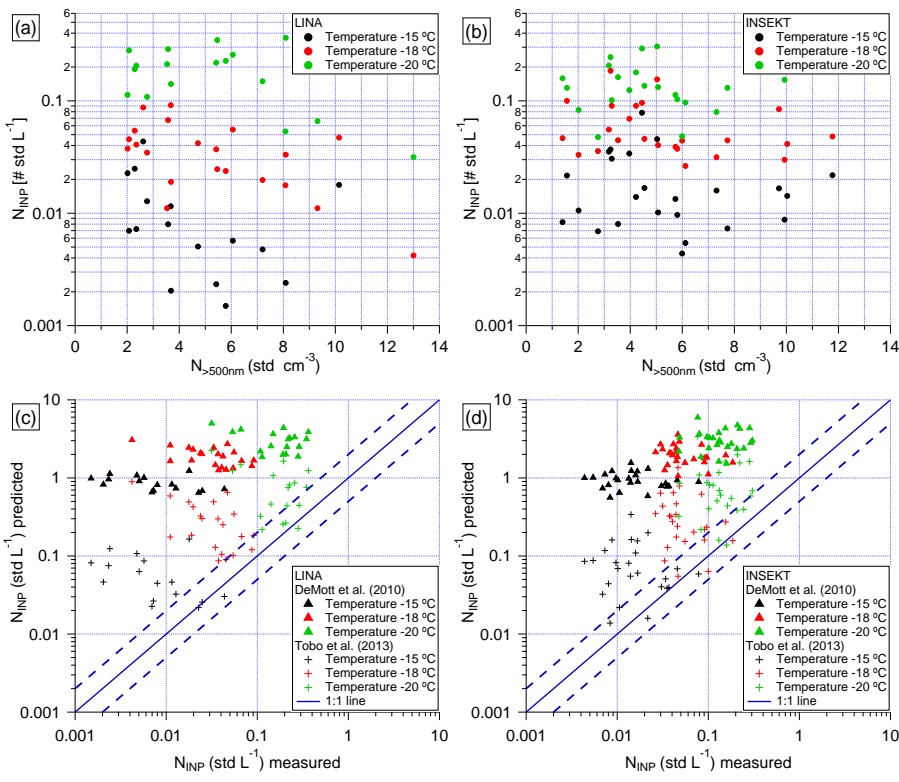

**Figure 12.** Scatter plot of $N_{INP}$ measured by LINA (a) and INSEKT (b) against $N_{>500nm}$. Scatter plot of $N_{INP}$ measured by LINA (c) and INSETK (d) against the $N_{INP}$ predicted by DeMott et al. (2010) and Tobo et al. (2013). The dashed lines outline a range of a factor of 2 about the 1:1 line (solid line).





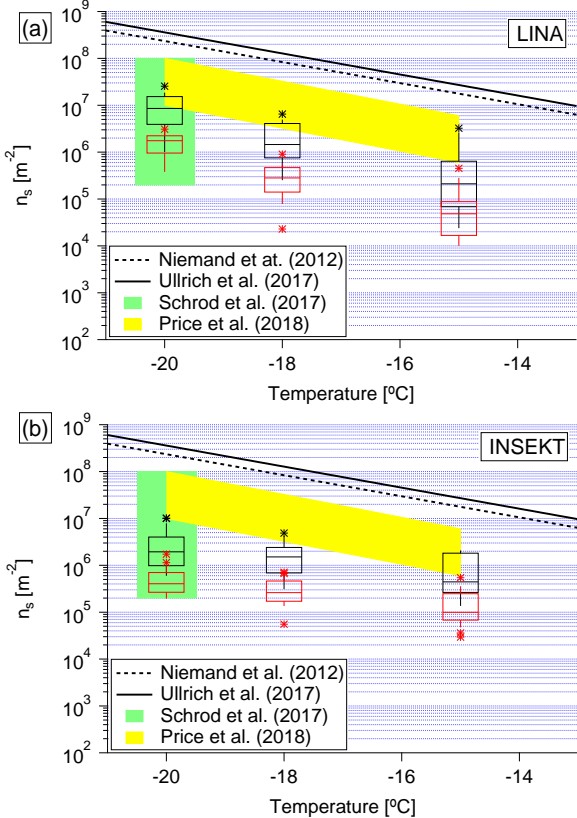

**Figure 13.** $n_{s>500\,nm}$ (black box plot) and $n_{s\_all}$ (red box plot) as a function of temperature. The results were determined based on LINA-measured $N_{INP}$ in Fig.(a) and INSEKT-measured $N_{INP}$ in Fig.(b). The boxes represent the interquartile range. Data not included between the whiskers are plotted as an outlier with a star. Two $n_s$ parameterizations (Niemand et al., 2012; Ullrich et al., 2017) for desert dust are shown in dashed and solid line. We also compare to recent data from airborne measurement by Schrod et al. (2017) and Price et al. (2018), as shown in green and yellow shadow, respectively.