# Peer review of "Characterization of aerosol properties at Cyprus, focusing on cloud condensation nuclei and ice nucleating particles"

_Atmospheric Chemistry and Physics, 2019_

## Referee Comment (RC1) · Anonymous Referee #2 · 4 Jul 2019

Review of "Characterization of aerosol properties at Cyprus, focusing on cloud condensation nuclei and ice nucleating particles" by Gong et al.

This manuscript focuses on the abundance, properties, and sources of CCN and INP on the island of Cyprus. Measurements at Cyprus are unique from other regions since Cyprus can be influenced by a range of different and complex aerosol mixtures, including mineral dust from the Sahara Desert and anthropogenic aerosols from Europe and the Middle East. From the CCN measurements the authors conclude the Aiken mode aerosols contain organics and the accumulation mode aerosols contain sulfate. On the topic of INP, the authors suggest the INP are mainly from long range transport with

a few samples influenced by biological INP from Cyprus. A parameterization based on N>500 nm overestimate INP by 1-2 orders of magnitude, and measured ns values were much lower than ns values for mineral dust.

This study adds to the growing body of data on the abundance, properties, and sources of CCN and INP in the atmosphere. The data analysis in this manuscript is especially impressive. However, I felt that a few of the conclusions were too strong or not well supported. Below are comments the authors should address before publication:

1. When discussing the INP data (Abstract and Section 3.4) the authors refer to the aerosol particles at Cyprus as "anthropogenically polluted aerosols". This gives the impression that anthropogenic aerosol dominated during the campaign. On the other hand, in the Introduction and Experimental they discuss both natural and anthropogenic sources for Cyprus, and their results show relatively low NOx concentrations during most of the campaign. Based on this, is "anthropogenically polluted aerosols" the best description of aerosols at Cyprus during the measurements.

2. Page 10, lines 25-30 and Abstract. The authors suggest the presence of sulfate in the accumulation mode aerosols based on a median kappa value of 0.57. Could a Kappa of 0.57 also be explained by a mixture of sodium chloride and organics? Please discuss if there are other possible explanations of kappa = 0.57.

3. Page 8, line 15-18. Change "less than 5% of the trajectories" to "approximately 5% of the trajectories" or something similar since less than 5% could be 0%.

4. Page 8, line 25-26. Change "the corresponding air masses originating from the Sahara Desert or the desert regions in Syria and Iraq" to "the corresponding air masses originating from dust areas" to be consistent with what is shown in Figure 5.

5. Page 11, lines 20. "These observations are indicative for the absence of nearby sources, and hence we conclude that the sampling INP, at least those ice active at < -15 C, originate from long range transport." I do not think this conclusion is well supported

since Fig S9 is also consistent with similar concentrations of INP from nearby land and ocean.

6. Page 12, line 20-22. Is a log-normally distributed Ninp population proof that the INP originated from long range transport rather than local sources? This seems like too strong of a statement, since it implies that the only mechanism capable of producing a log-normally distributed Ninp population is long range transport. Welti et al. 2018 showed that a lognormal distribution can be explained by random dilution during transport, but did they show that this is the only mechanism capable of forming a log-normally distributed population? Please discuss in the manuscript.

---

## Referee Comment (RC2) · Anonymous Referee #1 · 20 Jul 2019

This study reported the new particle formation, aerosol hygroscopicity, and ice nucleation activities in the atmosphere at Cyprus. The manuscript is well-written and very clear. But, there are several questions should be addressed in the revised version. (1) In conclusions, the author mentioned "frequently NPF and growth events were observed". As described in the text, most of bursts in nucleation particles attribute to airport emissions, but not NPF. The wording "frequently" may not be properly. (2) The samples were separated into "ocean" and "land" samples. How about the effects of "land and sea breeze" on the samples? The "land" air may blow to the ocean, and later will come back again. This may explain why the "ocean" samples is similar to that of "land". (3) In the abstract, "with a median $\kappa$ value of 0.57, suggesting the presence of

sulfate.". Actually, 0.57 means almost of pure sulfate. The sea salt can also go down to accumulation mode particles. A high k value may indicate the presence of sea salt. (4) In page 6, Line 10: "Each filter was immersed into 1 mL ultrapure water". 1 ml is enough to wash the particle off from the filter? (5) Typically, the particle surface areas concentration is calculated assuming a spherical shape. While, dust particle may be more irregular, as a result, lead to increase in the surface areas.

---

## Author Comment (AC1) · 30 Jul 2019

Dear Reviewer,

We thank you for doing this review and for your suggestions that helped to improve our manuscript. Below, please find your original comments in blue and our responses in black. When referencing page and line numbers, we are always referring to the original versions of manuscript and SI.

This manuscript focuses on the abundance, properties, and sources of CCN and INP on the island of Cyprus. Measurements at Cyprus are unique from other regions since Cyprus can be influenced by a range of different and complex aerosol mixtures, including mineral dust from the Sahara Desert and anthropogenic aerosols from Europe and the Middle East. From the CCN measurements the authors conclude the Aiken mode aerosols contain organics and the accumulation mode aerosols contain sulfate. On the topic of INP, the authors suggest the INP are mainly from long range transport with a few samples influenced by biological INP from Cyprus. A parameterization based on N>500 nm overestimate INP by 1-2 orders of magnitude, and measured ns values were much lower than ns values for mineral dust.

This study adds to the growing body of data on the abundance, properties, and sources of CCN and INP in the atmosphere. The data analysis in this manuscript is especially impressive. However, I felt that a few of the conclusions were too strong or not well supported. Below are comments the authors should address before publication:

1. When discussing the INP data (Abstract and Section 3.4) the authors refer to the aerosol particles at Cyprus as "anthropogenically polluted aerosols". This gives the impression that anthropogenic aerosol dominated during the campaign. On the other hand, in the Introduction and Experimental they discuss both natural and anthropogenic sources for Cyprus, and their results show relatively low NOx concentrations during most of the campaign. Based on this, is "anthropogenically polluted aerosols" the best description of aerosols at Cyprus during the measurements.

You are correct in that the aerosol likely has many sources, and that the rather low NOx concentrations show that this is not a typical anthropogenically polluted aerosol. We've changed to wording to "polluted aerosol of the eastern Mediterranean" in the abstract, in Sec. 3.4 and in the conclusions.

2. Page 10, lines 25-30 and Abstract. The authors suggest the presence of sulfate in the accumulation mode aerosols based on a median kappa value of 0.57. Could a Kappa of 0.57 also be explained by a mixture of sodium chloride and organics? Please discuss if there are other possible explanations of kappa = 0.57.

Following one of the second reviewer's remarks, we extended the discussion in lines 25- 30, page 10 and changed the abstract and conclusion accordingly.

 "A few sea salt particles mixed with organic carbon might also be present in the accumulation mode, according to a previous study (Prather et al., 2013). But the absolute number concentration of sea salt mixed with organic carbon particle in the size range <200 nm is likely limited."

3. Page 8, line 15-18. Change "less than 5% of the trajectories" to "approximately 5% of the trajectories" or something similar since less than 5% could be 0%.

Done.

4. Page 8, line 25-26. Change "the corresponding air masses originating from the Sahara Desert or the desert regions in Syria and Iraq" to "the corresponding air masses originating from dust areas" to be consistent with what is shown in Figure 5.

Done.

5. Page 11, lines 20. "These observations are indicative for the absence of nearby sources, and hence we conclude that the sampling INP, at least those ice active at < -15C, originate from long range transport." I do not think this conclusion is well supported since Fig S9 is also consistent with similar concentrations of INP from nearby land and ocean.

We followed your recommendation, and deleted the following lines 20-21, page 11: "These observations are indicative for the absence of nearby sources, and hence we conclude that the sampled INP, at least those ice active at -15 °C, originate from long-range transport."

Following one of the second reviewer's remarks, we extended the discussion here.

"A source apportionment for INP examined in this study is therefore difficult to do. Considering that Cyprus is only a small island surrounded by ocean, its effect might be limited. Besides, for a location such as Cyprus, it is difficult to determine sources for different air masses only based on wind direction, alone."

6. Page 12, line 20-22. Is a log-normally distributed Ninp population proof that the INP originated from long range transport rather than local sources? This seems like too strong of a statement, since it implies that the only mechanism capable of producing a log-normally distributed Ninp population is long range transport. Welti et al. 2018 showed that a lognormal distribution can be explained by random dilution during transport, but did they show that this is the only mechanism capable of forming a log-normally distributed population? Please discuss in the manuscript.

In addition to what you write in your comment, Welti et al. (2018) explained that the unimodal, regular lognormal shape of the frequency distribution (or PDF) indicates the absence of a strong local source. If there is a nearby source, the distribution will be skewed with a stronger downward bend at high concentrations. Welti et al. (2018) refer to Ott (1990), where this behavior was already explained. We therefore feel that the statement we make is not too strong. We did, however, include the following information at the end of the sentence you refer to:

[revised manuscript text omitted]

When data evaluation was started for this set of samples, tests were made (as these were the first atmospheric samples on polycarbonate for which we did an analysis). A set of measurements was done in which filters were washed off with 1 mL of ultrapure water, first. This was done by shaking the centrifuge tube in which filter and water were situated. From this, 0.1 mL was used for a first analysis, directly taken from the tube in which the shaking had been done. Then 9.1 mL were added to the tube and the sample was shaken again, followed by a second analysis. The results from both dilutions can be seen in Fig. S3, and data in the overlapping temperature region are well in agreement. Based on this, we decided to use only 1 mL for washing, as this allows us to retrieve INP concentrations already at higher temperatures.

[Figure]

**Figure S3.** $N_{INP}$ measured by LINA as a function of temperature. The solid triangles and hollow circles show $N_{INP}$ from the samples washed with 1 mL and 10 mL ultrapure water, respectively.

**S4   Wind speed and direction**

[revised manuscript text omitted]

---

## Author Comment (AC2) · 30 Jul 2019

Dear Reviewer,

We thank you for doing this review and for your suggestions that helped to improve our manuscript. Below, please find your original comments in blue and our responses in black. When referencing page and line numbers, we are always referring to the original versions of manuscript and SI.

This study reported the new particle formation, aerosol hygroscopicity, and ice nucleation activities in the atmosphere at Cyprus. The manuscript is well-written and very clear. But, there are several questions should be addressed in the revised version.

(1) In conclusions, the author mentioned "frequently NPF and growth events were observed". As described in the text, most of bursts in nucleation particles attribute to airport emissions, but not NPF. The wording "frequently" may not be properly.

We followed your recommendation, removed the "frequently" in this sentence. In this study, we only focused on 3 NPF events, since particles grown to this sizes makes them potential CCN. There are ~2 more NPF events that were observed in sub-10 nm but during these, particles did not grow to larger sizes. There will be an upcoming paper to discuss the NPF in the sub-10nm size range which will also compare with the data in this study.

However, here we would like to stress that the pollution from the airport was filtered out when we analyzed NPF events and CCN number concentration (as we say in 3.2: "Therefore, in the following, time periods affected by pollution from the airport were excluded from further analysis.").

(2) The samples were separated into "ocean" and "land" samples. How about the effects of "land and sea breeze" on the samples? The "land" air may blow to the ocean, and later will come back again. This may explain why the "ocean" samples is similar to that of "land".

We followed your recommendation, and extended the discussion after line 21, page 11.

"A source apportionment for INP examined in this study is therefore difficult to do. Considering that Cyprus is only a small island surrounded by ocean, its effect might be limited. Besides, for a

location such as Cyprus, it is difficult to determine sources for different air masses only based on wind direction, alone."

(3) In the abstract, "with a median κ value of 0.57, suggesting the presence of sulfate.". Actually, 0.57 means almost of pure sulfate. The sea salt can also go down to accumulation mode particles. A high k value may indicate the presence of sea salt.

Following one of the second reviewer's remarks, we extended the discussion in lines 25- 30, page 10 and changed the abstract and conclusion accordingly.

"A few sea salt particles mixed with organic carbon might also be present in the accumulation mode, according to a previous study (Prather et al., 2013). But the absolute number concentration of sea salt mixed with organic carbon particle in the size range <200 nm is likely limited."

(4) In page 6, Line 10: "Each filter was immersed into 1 mL ultrapure water". 1 ml is enough to wash the particle off from the filter?

Obviously, $N_{INP}$ from LINA (1 mL ultrapure water washed) agreed well with INSEKT (8 mL ultrapure water washed). Therefore, we think 1 mL ultrapure water is enough to wash off all particles.

[Figure]

Figure S3. $N_{INP}$ measured by LINA as a function of temperature. The solid triangles and hollow circles show $N_{INP}$ from the samples washed with 1 mL and 10 mL ultrapure water, respectively.

When data evaluation was started for this set of samples, tests were made (as these were the first atmospheric samples on polycarbonate for which we did an analysis). A set of measurements was done in which filters were washed off with 1 mL of ultrapure water, first. This was done by shaking the centrifuge tube in which filter and water were situated. From this, 0.1 mL was used for a first analysis, directly taken from the tube in which the shaking had been done. Then 9.1 mL were added to the tube and the sample was shaken again, followed by a second analysis. The results from both dilutions can be seen in Fig. S3, and data in the overlapping temperature region are well in agreement. Based on this, we decided to use only 1 mL for washing, as this allows us to retrieve INP concentrations already at higher temperatures.

We added the above text and the figure to the SI, and the text below to the main text:

"It should be mentioned that results from separate tests using 1 mL and 10 mL of washing water were well in agreement (see Fig. S3)."

(5) Typically, the particle surface areas concentration is calculated assuming a spherical shape. While, dust particle may be more irregular, as a result, lead to increase in the surface areas.

We calculated the particle surface area concentrations by assuming a spherical shape. Actually, we considered the particle shape when we corrected the MPSS measured number concentration for multiple charged particles in the APS size range. The electrical mobility diameter (measured by MPSS) and aerodynamic diameter (measured by APS) were converted to the volume equivalent particle diameter.

The dry dynamic shape factor $\chi$ of mineral dust is $\chi = 1.25$ (Kaaden et al., 2009) for 1 µm particles, whereas the dynamic shape factor for sodium chloride is $\chi = 1.08$ (Kelly and McMurry, 1992;Gysel et al., 2002). We used the average shape factor of 1.17 in this study. We added this information in the Section 2.2.

[revised manuscript text omitted]

When data evaluation was started for this set of samples, tests were made (as these were the first atmospheric samples on polycarbonate for which we did an analysis). A set of measurements was done in which filters were washed off with 1 mL of ultrapure water, first. This was done by shaking the centrifuge tube in which filter and water were situated. From this, 0.1 mL was used for a first analysis, directly taken from the tube in which the shaking had been done. Then 9.1 mL were added to the tube and the sample was shaken again, followed by a second analysis. The results from both dilutions can be seen in Fig. S3, and data in the overlapping temperature region are well in agreement. Based on this, we decided to use only 1 mL for washing, as this allows us to retrieve INP concentrations already at higher temperatures.

[Figure]

**Figure S3.** $N_{\text{INP}}$ measured by LINA as a function of temperature. The solid triangles and hollow circles show $N_{\text{INP}}$ from the samples washed with 1 mL and 10 mL ultrapure water, respectively.

**S4    Wind speed and direction**

[revised manuscript text omitted]